# Measuring the Complex Permittivities of Plastics in Irregular Shapes

**DOI:** 10.3390/polym13162658

**Published:** 2021-08-10

**Authors:** Hsien-Wen Chao, Hua-Hsuan Chen, Tsun-Hsu Chang

**Affiliations:** Department of Physics, National Tsing Hua University, 101, Section 2, Kuang Fu Road, Hsinchu 300044, Taiwan; s9822817@m98.nthu.edu.tw (H.-W.C.); st9820342@yahoo.com.tw (H.-H.C.)

**Keywords:** complex permittivity, enhanced-field method, irregularly shaped, contour mapping method

## Abstract

This work presents the measurement of the complex permittivities of high density polyethylene (HDPE), linear low density polyethylene (LLDPE), low density polyethylene (LDPE), polypropylene (PP), Nylon, and thermoplastic vulcanizates (TPV) in irregular shapes at the microwave frequency. A Teflon sample holder was employed to pack irregularly shaped plastic materials with various volumetric percentages. The samples were put into a resonant cavity with an enhanced electric field in its center, which is known as the enhanced-field method (EFM). The resonant frequencies and the quality factors at different volumetric percentages were measured by a network analyzer and compared with simulated results using a full-wave simulator (high-frequency structure simulator (HFSS)). Three simulation models, layer, ring, and hybrid, are proposed and compared with the experimental results. It is found that the hybrid model (denoted as Z5R5) with five heights and five radii in the partition is the most suitable. The complex permittivities of six plastic materials were evaluated by the contour maps of the HFSS simulation using the hybrid model. The measured complex permittivities of the irregularly shaped polymers agree well with their counterparts in bulk form.

## 1. Introduction

Plastics have been widely used in various fields, such as household goods, packaging materials, food industry, manufacturing, electronic devices, communications, and the aviation industry. Although plastic raw materials are relatively cheap, recycling plastic materials is a very important issue for global environmental protection. Recyclable plastics come in many types. Among them, polyethylene (PE), polypropylene (PP), Nylon, and thermoplastic vulcanizates (TPV) are studied. The most common form of low density polyethylene (LDPE) is film-like and can be used in microwave ovens. The linear low density polyethylene (LLDPE) is a blended form of LDPE. The molecules of LLDPE all lineup and strongly hold together as the film is stretched to achieve much more flexibility, tensile strength, and conformability than LDPE. The high density polyethylene (HDPE) is better than those of LDPE and LLDPE in toughness, strength, and chemical resistance, but the flexibility of HDPE is not good. Polypropylene (PP) is a thermoplastic polymer used in a wide variety of applications, such as laboratory equipment, automotive parts, medical devices, and food containers. PP is produced via chain-growth polymerization from the monomer propylene. The characteristics of Nylon contain high wear resistance, excellent heat resistance, excellent chemical resistance, ease of machine, and noise dampening. TPV (thermoplastic vulcanizates) is a kind of thermoplastic rubber (TPR), which possesses the thermal resistance and low compressive deformation of vulcanized rubber, and the high-temperature resistance can reach 135 °C. These plastic materials can be applied in the field of 5G communication, but they must be low-loss. Characterizing their complex permittivities is essential.

The measurement of complex permittivities can be classified into several methods, such as the coaxial probe [1,2,3,4], the transmission line [5,6], the free space [7,8,9,10], the parallel plate [11,12], and the resonant cavity [13,14,15]. The coaxial probe method is applied for broadband, non-destructive measurement, and good for lossy materials. It is suitable for liquids and semi-solids, such as honey, gel, and cream. The transmission line method is also a broadband technique and good for the low-loss to lossy materials. It is suitable for machinable solids and requires a sample of specific dimensions. The free space method is used in the broadband, and is best for non-contacting, flat sheets. It can be used in the continuing heat of high-temperature. The resonant cavity method (e.g., the perturbation method) is accurate but with a narrow bandwidth and is suitable for low-loss materials. It requires a sample of specific dimensions and its size must be much smaller than that of the resonant cavity. The parallel plate method is accurate and best for low frequencies, thin, and flat sheets. However, not all raw materials can be easily manufactured into a specified dimension or shape samples of the above-mentioned measurement methods. For example, the raw materials of plastics may come in granules. In addition, fibrous polymer materials are also difficult to measure [16,17,18].

To solve the above problems, this report proposes a method for the measurement of the complex permittivity of irregularly shaped materials. In this study, plate-shaped plastic materials were cut into strips and generated as irregularly shaped. The irregularly shaped materials with different volumetric ratios can be crammed into a Teflon holder. The complex permittivities with different volumetric percentages will be used to extrapolate the complex permittivity of irregularly shaped material at 100% volumetric percentage. Six plastics, HDPE, LLDPE, LDPE, PP, Nylon, and TPV, will be examined. The correctness of the proposed method will be validated with the bulk samples. The measured complex permittivities might provide useful information for understanding the chemical structures of polymers.

## 2. Measured Approach and Procedure

The polymer (plastic) in irregular shape can be obtained from its bulk form. Since the irregularly-shaped sample and the bulk sample are exactly the same material, their complex permittivity should be the same. This allows us to validate the correctness of the proposed method. The six pieces of plastic from the Plastic Industry Development Center (PIDC) of Taichung, Taiwan were cut into strips, and then the plastic strips were placed in a pencil sharpener, and the strips were cut into irregular shapes. The more uneven the shape of the material, the better the uniformity we can achieve. Figure 1 illustrates the sample preparation procedures, which are simple, low cost, and time-saving.

Figure 2a shows the experimental setup. A rounded rod is designed in the center of the resonant cavity and the sample is put at the top of a rounded rod [19]. The output signal (1 mW) is transmitted from the SMA (SubMiniature version A) port of the Anritsu MS46122A network analyzer to the SMA port of the resonant cavity through a coaxial cable. The reflected signal is monitored in the computer to evaluate the resonant frequency and *Q*-factor of the sample. The bulk material is placed inside the Teflon holder and sealed by the top cover of Teflon shown in Figure 2b. The volume of bulk is the same as the filling volume of the Teflon holder and will be regarded as a 100% volume percentage of the measured sample size. The irregularly shaped materials are crammed into the Teflon holder from 10% to less than 50% volumetric percentages of bulk and are shown in Figure 2c. The external dimensions of the Teflon holder are 11 mm in radius and 8.75 mm in height. Its internal dimensions are 6.9 mm in radius and 6.7 mm in height.

The resonant frequency and the *Q*-factor of the material can be determined by the network analyzer. However, the resonant frequency is a function of the complex permittivity, so is the *Q*-factor [20]. We can establish the contour map spanned by the resonant frequency and the *Q*-factor using the HFSS simulation. The contour map can be used to uniquely determine the complex permittivity of a material [21]. The cross-sectional structure of the resonant cavity is shown in Figure 3a, which includes an SMA port, a brass main cavity, a brass top cover, a Teflon top cover, and a Teflon holder. The sample is filled inside the Teflon holder and sealed by the top cover of Teflon. The input and output of the signal can be transmitted through the SMA port. For example, the dielectric constants from 2 to 4 and loss tangents from 0 to 0.1 input the HFSS model of the resonant cavity in Figure 3b. The contour map of the resonant frequencies and the *Q*-factors can be established in Figure 4.

Figure 4 can be used to determine the complex permittivity from the measured parameters. For example, if the measured resonant frequency and *Q*-factor of an unknown material are 2.3929 GHz and 295.8. The resonant contour line of 2.3929 GHz (blue) and the *Q*-factor contour line (red) of 295.8 will intersect at one and only one point. This point is related to 3.2 on the horizontal axis (the relative dielectric constant) and 0.0355 on the vertical axis (the loss tangent). In short, the relative dielectric constant and the loss tangent of the material are 3.2 and 0.0355.

## 3. Specimen Preparation and Analysis

The molecular structures of HDPE, LLDPE, and LDPE are all based on PE, but their arrangements are different. The molecular structure of PP is similar to that of PE, while the molecular structure of Nylon is different from that of PE. TPV is not only a mixture, but its molecular structure is very different from that of PE. The authors want to clarify and study the relationship between the molecular structures and the complex permittivity of the materials. This is why we chose the six materials. The plate-plastic of HDPE, LLDPE, LDPE, PP, Nylon, and TPV were fabricated in the specified samples with a height of 6.7 mm and a radius of 6.9 mm. Of course, the bulk sample preparation process was difficult and required a high processing technique. The specified samples were placed inside the Teflon holder and sealed using the top cover of Teflon, respectively. The resonant frequencies and *Q*-factors of bulks of HDPE, LLDPE, LDPE, PP, Nylon, and TPV were measured and monitored on the computer. The dielectric constants (experiment/reference) and loss tangents (experiment/reference) of HDPE, LLDPE, LDPE, PP, Nylon, and TPV of bulks were shown in Table 1. In six plastic bulks, the experimental results are almost the same as the references. In other words, the enhanced electric field (E-field) method [19,20] of the resonant cavity and the contour mapping method [21] we proposed are very reliable.

It can be clearly observed in Figure 5a,b that the dielectric constants and loss tangents of HDPE, LLDPE, and LDPE distribute very closely since they are the same series of PE products. The molecular structures of HDPE, LLDPE, and LDPE are illustrated in Figure 6a–c and their main structures are the same. Their complex permittivities may be related to the main structures of the molecular structure. Similarly, in Figure 5a,b, the dielectric constant and loss tangent of PP distributes very closely with those of HDPE, LLDPE, and LDPE.

We can find that the molecular structures of PE and PP are very close, as shown in Figure 6a–d. The molecular structures of Nylon and TPV shown in Figure 6e,f are very different from those of PP and PE products. Therefore, the dielectric constant and loss tangent of nylon and TPV are distinct from those of PP and PE products. The PP and PE products are suitable for the 5G communication due to their lower loss tangents.

## 4. Comparison with Layer, Ring, and Hybrid Models

In order to measure the complex permittivity of the irregularly shaped material of Nylon, first, a layer model was proposed. The irregularly shaped materials of different volumetric ratios would be filled respectively in the Teflon holder and measured the resonant frequencies and *Q*-factors. In Figure 7a, the layer number from 3 to 6 in the layer models with different volumetric ratios were simulated by the HFSS. The resonant frequencies and the reciprocal of *Q*-factors of different layer models related to the volumetric ratios were shown in Figure 7b,c. The layer models of different layers do not match the experimental results. It is found that the dielectric constants from these layer models were overestimated, and the loss tangents were underestimated compared with the experimental results. In Figure 7c, the reciprocal of *Q*-factor is always employed rather than the *Q*-factor due to the simple proportional relation between the reciprocal of *Q*-factor and loss tangent, thus we are trendy to employ the reciprocal of *Q*-factor.

Since the results of the layer models could not match the experimental results, a ring model was then proposed. The ring number from 3 to 6 in the ring model is shown in Figure 8a. Figure 8b,c showed the dielectric constants and the reciprocal of *Q*-factors of different ring models from HFSS simulation related to the volumetric ratios. The results of different ring models and experiments still did not match, either. The trends of the ring models compared with those of the layer models were opposite. Compared with the experimental results, the dielectric constants of these layer models were underestimated, and the loss tangents were overestimated.

Although the results of the layer and ring model we proposed could not match with the experimental results. Fortunately, both models could be complementary to each other. Due to the characteristics of the layer model and the ring model, one is overestimated and the other is underestimated. A combination of the layer model and ring model was produced and called a hybrid model, of which the structure is shown in Figure 9a. The signs for the hybrid model were called ZmRn. The signs of Z and R were respectively defined as z-direction and radius direction, and m and n are from 4 to 7. Compared with the dielectric constant of LLDPE bulk in Table 1, Figure 9b–e showed the errors of dielectric constants related to the different volumetric percentages in the models of Z4R4, Z5R5, Z6R6, and Z7R7. The errors of Z5R5 with different volumetric percentages shown in Figure 9c were the smallest compared with the experimental results. Similarly, the errors of loss tangents related to the different volumetric percentages of the Z5R5 were the smallest, as well. Therefore, the hybrid model of Z5R5 would be employed to evaluate the complex permittivities of HDPE, LLDPE, LDPE, PP, TPV, and Nylon, which would be compared with Table 1.

Certainly, the shredded-shape (irregular shape) samples with the different volumetric percentages can be extrapolated to the complex permittivity of 100% volumetric percentage (regarded as a bulk). From this viewpoint, the establishment of a simulation model seems unnecessary. Figure 9a–d shows that the higher the filling ratio of volumetric percentage, the more accurate the simulated results of the models related to the measurement results of bulks. This point is very important. It can show that we only need to fabricate and measure a sample with the highest filling ratio of volumetric percentage rather than samples with different filling ratios of volumetric percentages. It is certainly more time-saving and cost-effective to fabricate and measure just one sample.

The foundation of the above mentioned simplified technique is based on the reliability of the model. In this experiment, the irregular shape materials are randomly crammed into and taken out of the Teflon holder five times for each volumetric percentage. The shredded shape and size distribution of crammed materials are different each time with a fixed weight. In Figure 9a–d, the standard deviations of the dielectric constant related to the volume percentage in the Z4R4, Z5R5, Z6R6, and Z7R7 models were observed. Their standard deviations are all less than 1%. In other words, the shredded shape and size distribution of crammed materials are seemingly independent of the experimental setups. In addition, this method can be extended to powder and fiber types. It is well-known that the characteristics of materials under the nanoscale are very different from those above the nanoscale. In this study, the irregularly-shaped samples are relatively large. The nanoscale effect which strongly depends on the volumetric percentage can be neglected.

Figure 10a–f showed the comparisons of the layer, ring, and hybrid models and experiments for six plastics. The trends of the different volumetric percentages could be extrapolated to the resonant frequency and the reciprocal of *Q*-factor of 100% volumetric percentage. The trends of Figure 10a–f illustrated that the hybrid model of Z5R5 was the most matched with the experiment. The extrapolated values (100% volumetric percentage) of the resonant frequencies and the reciprocal of *Q*-factors could evaluate the dielectric constants and loss tangents of plastic materials by the contour maps.

Table 2 shows the dielectric constants and loss tangents of six plastics in the bulks and the Z5R5 models. Except that the loss tangents of Nylon obtained from the bulk and Z5R5 models were a little distinct, and the other results are almost the same.

It is interesting to know why the hybrid model is superior to the layer and ring models. In a real condition, the directions of the electric fields can be divided into vertical and parallel components to the surfaces of the irregularly shaped materials. However, the layer/ring model only considers the vertical/parallel component of the E-field to the surface of the irregularly shaped materials. The layer model consists of several discs, each of which has a small aspect ratio (height divided by diameter). The directions of the E-field in the cavity are almost the z-components [20], thus the surfaces of the discs are almost vertical to that of the E-field. According to the boundary condition on the electromagnetic field, it can be written as follows:(1)$${\vec{D}}_{1n}={\vec{D}}_{2n}\Rightarrow \varepsilon_{1}{\vec{E}}_{1n}=\varepsilon_{2}{\vec{E}}_{2n}\Rightarrow \;\varepsilon_{1}E_{1z}=\varepsilon_{2}E_{2z}\;\Rightarrow \;E_{2z}=\frac{\varepsilon_{1}}{\varepsilon_{2}}E_{1z}$$
where 1 is air, and 2 is the medium. The power dissipation formula of the dielectric loss in the medium can be written as follows:(2)$$P_{d}=\frac{1}{2}\omega \varepsilon_{2}^{\prime \prime }{\int \left|\frac{\varepsilon_{1}^{\prime }}{\varepsilon_{2}^{\prime }}E_{1z}\right|}^{2}ds$$
*P_d_* is dissipated power, ω is angular frequency, $\varepsilon_{2}^{\prime \prime }$ is an imaginary part of permittivity of material, and *E* is an electric field.

The ring model consists of several rings, each of which has a large aspect ratio (the height of the ring divided by the thickness of the ring). The directions of the E-field in the cavity are almost parallel to the surface of the rings. According to the boundary condition on the electromagnetic field, it can be written as follows:(3)$${\vec{E}}_{1t}={\vec{E}}_{2t}\Rightarrow E_{2z}=E_{1z}$$

The power dissipation formula of the dielectric loss in the medium can be written as follows:(4)$$P_{d}=\frac{1}{2}\omega \varepsilon_{2}^{\prime \prime }{\int \left|E_{1z}\right|}^{2}ds$$

The layer model can predict that the dielectric constant and loss tangent are underestimated due to Equations (1) and (2). The dielectric constant of the air $\left(\varepsilon_{1}^{\prime }\right)$ is equal to 1 and that of the medium $\left(\varepsilon_{2}^{\prime }\right)$ is greater than 1, thus the dissipation factor $\varepsilon_{1}^{\prime }/\varepsilon_{2}^{\prime }$ is less than 1. The dissipated power of the layer model shown in Equation (2) will be smaller than that of irregularly shaped materials since the directions of the E-field are not only vertical to the surfaces of the irregularly shaped materials but also parallel to the surfaces of the irregularly shaped materials. Equivalently, in this layer model, the size of this material becomes smaller or its complex permittivity becomes smaller. For Equations (3) and (4), the dissipation factor of the ring model is equal to 1, and the dissipated power of the ring model will be greater than that of irregularly shaped materials. Just as the size of the material becomes larger or its complex permittivity becomes larger. In other words, the dissipation factor of irregularly shaped materials should be between $\varepsilon_{1}^{\prime }/\varepsilon_{2}^{\prime }$ and 1. Therefore, we try to employ a hybrid model that combines layer and ring models to fit the conditions in a real amorphous material.

The layer model underestimates the dielectric constant and loss tangent of irregularly shaped materials, while the ring model overestimates them. The resonant frequencies and loss tangents obtained using the hybrid model are closely related to the experimental results. The Z5R5 hybrid model seems to be the best one when considering the computing time and accuracy. In addition, if the filling ratio of an irregularly shaped material is higher, the simulated results using the hybrid model will be closer to the experimental results as expected. Even if the filling ratio is just 40%, the prediction of the Z5H5 model will be accurate enough.

## 5. Conclusions

This study proposed an approach to characterize the materials in irregular shapes. It can be used to measure the complex permittivities of polymer/dielectric row materials, powders, foods or small insects. The measured complex permittivities of HDPE, LLDPE, LDPE, and PP concentrate in a small region of the contour map due to the similar chemical structures. On the contrary, the chemical structures of Nylon and TPV are quite different, so are the complex permittivities. The proposed method can tell the difference of the chemical structure from the measured complex permittivities.

## Figures and Tables

**Figure 1 polymers-13-02658-f001:**
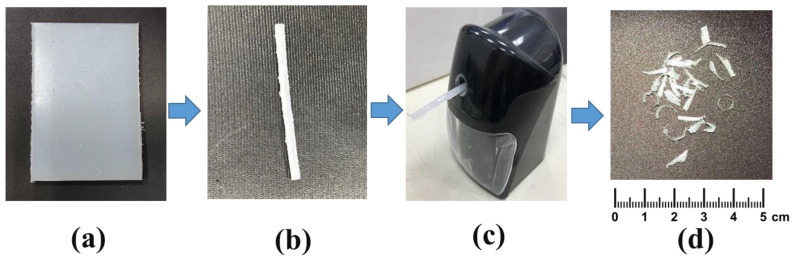
The procedures of sample preparation, (**a**) raw material, (**b**) strip, (**c**) pencil sharpener, and (**d**) irregular material.

**Figure 2 polymers-13-02658-f002:**
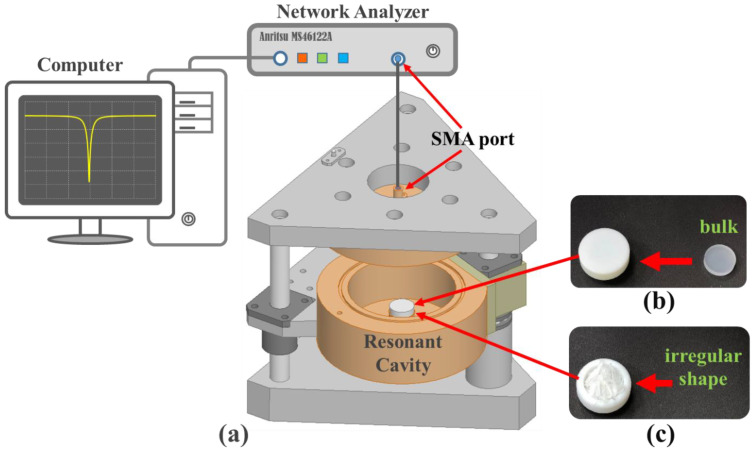
(**a**) The setup contains a network analyzer, a resonant cavity, computer, SMA connector, and cable, (**b**) the bulk material regarded as a reference sample of 100% volume percentage, (**c**) the irregularly shaped raw materials.

**Figure 3 polymers-13-02658-f003:**
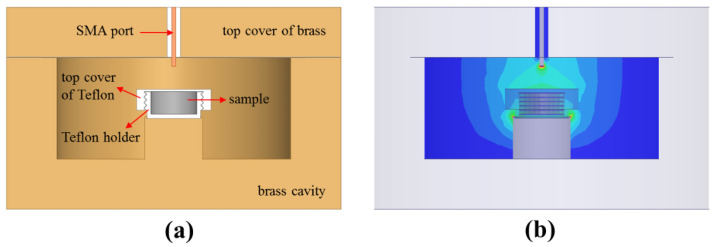
(**a**) The sectional structure of resonant cavity, and (**b**) the HFSS model of resonant cavity.

**Figure 4 polymers-13-02658-f004:**
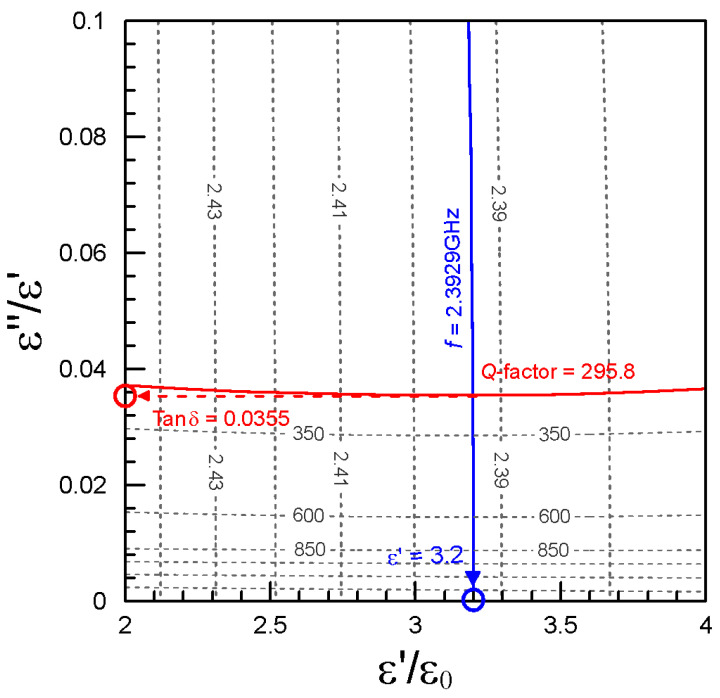
The contour map of resonant frequencies/loss tangents.

**Figure 5 polymers-13-02658-f005:**
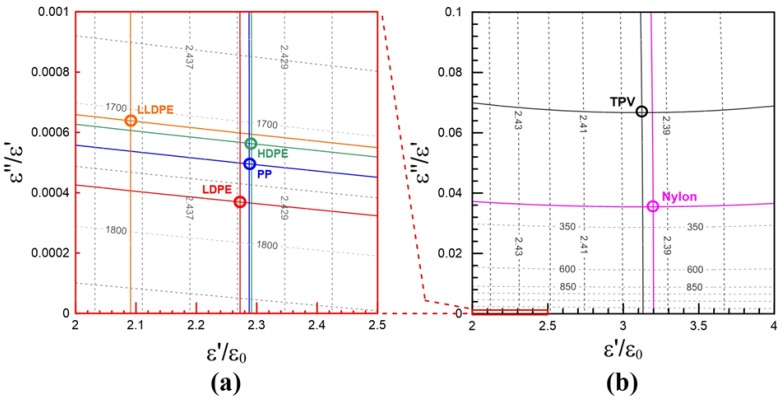
The dielectric constants and loss tangents of (**a**) HDPE, LLDPE, LDPE, PP, (**b**) Nylon, and TPV are determined by the contour mapping method.

**Figure 6 polymers-13-02658-f006:**
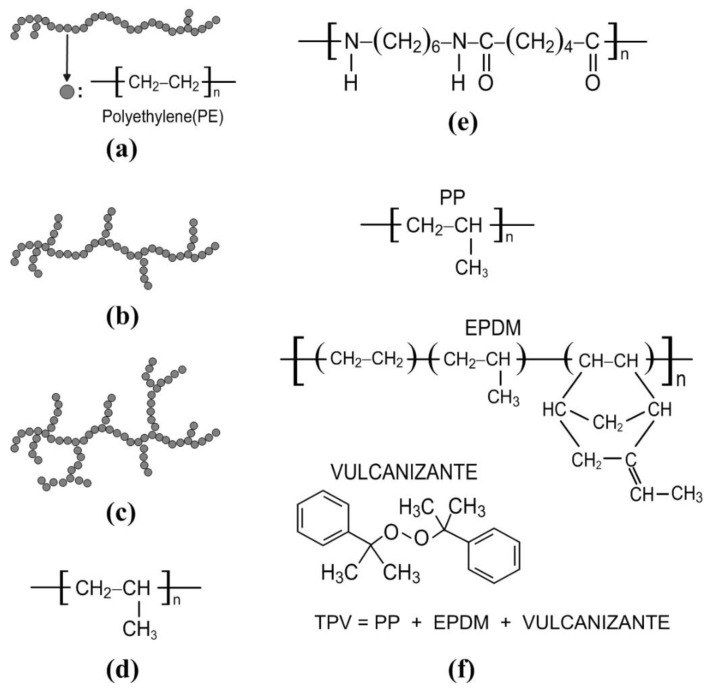
Molecular structures of (**a**) HDPE, (**b**) LLDPE, (**c**) LDPE, (**d**) PP, (**e**) Nylon, and (**f**) TPV.

**Figure 7 polymers-13-02658-f007:**
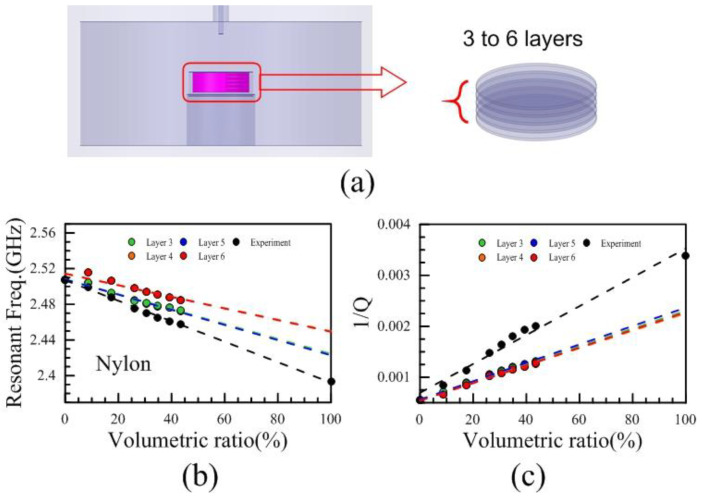
(**a**) The structure of layer model and layer number from 3 to 6, (**b**) the resonant frequencies related to different volumetric ratios of 3, 4, 5, 6 layer models compared with experimental results, (**c**) the reciprocal of *Q*-factor related to different volumetric ratios of 3, 4, 5, 6 layers models compared with experimental results.

**Figure 8 polymers-13-02658-f008:**
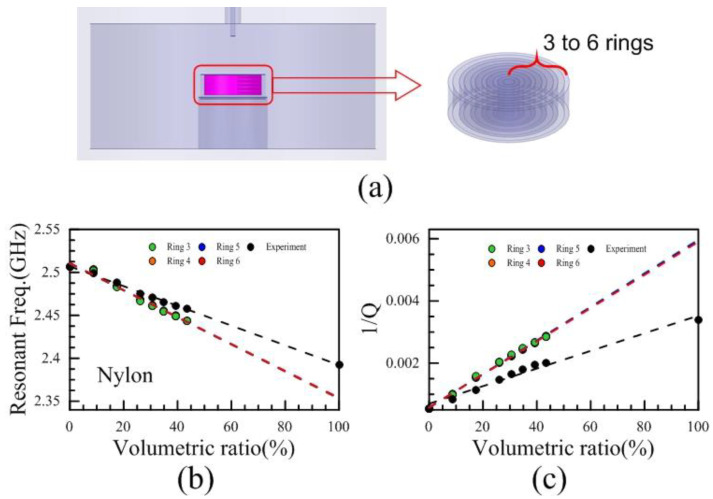
(**a**) The structure of ring model and ring number from 3 to 6, (**b**) the resonant frequencies related to the volumetric ratios of 3, 4, 5, 6 ring models compared with experimental results, (**c**) the reciprocal of *Q*-factors related to the volumetric ratios of 3, 4, 5, 6 ring models compared with experimental results.

**Figure 9 polymers-13-02658-f009:**
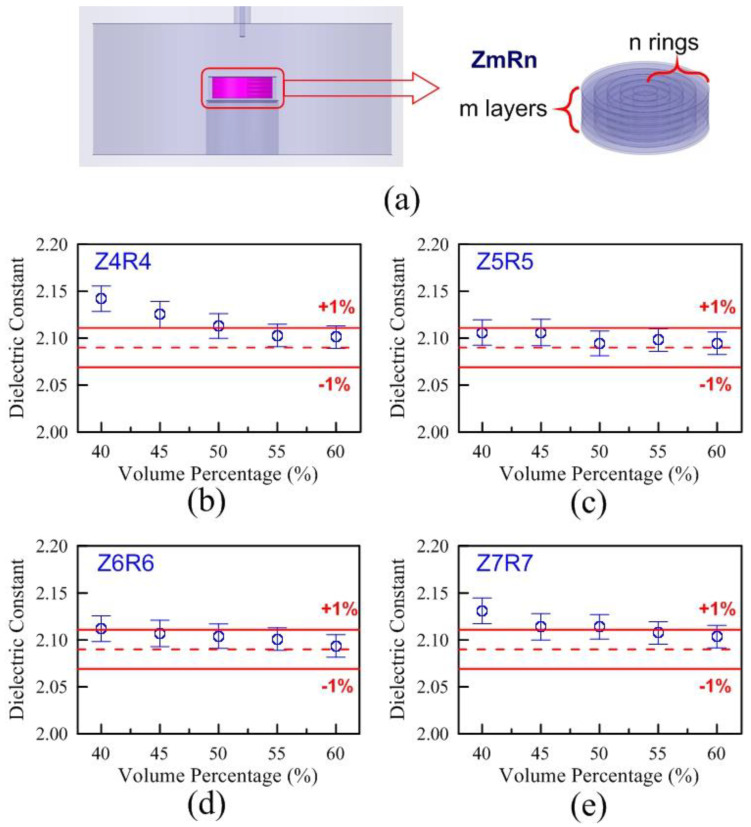
(**a**) There were m layers and n rings in the structure of the hybrid model. The errors of dielectric constants with volumetric ratios from the HFSS simulation in the (**b**) Z4R4, (**c**) Z5R5, (**d**) Z6R6, and (**e**) Z7R7 compared with the experimental results.

**Figure 10 polymers-13-02658-f010:**
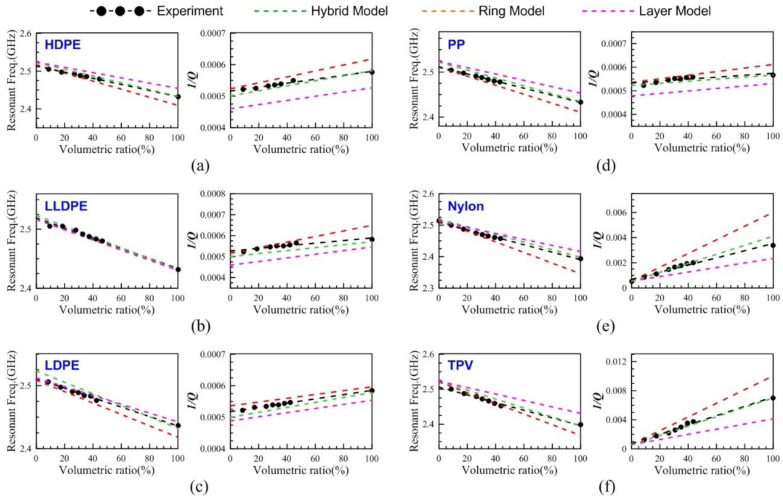
The resonant frequencies and the reciprocal of *Q*-factors of (**a**) HDPE, (**b**) LLDPE, (**c**) LDPE, (**d**) PP, (**e**) Nylon, and (**f**) TPV in the HFSS simulations of the hybrid model, ring model, layer model, and in the experiments related to the different volumetric ratios.

**Table 1 polymers-13-02658-t001:** The dielectric constants and loss tangents of six plastics.

	Dielectric Constant		Loss Tangent	
	Experiment (bulk)	References [22,23,24,25,26,27,28]	Experiment (bulk)	References [22,23,24,25,26,27,28]
HDPE	2.29	2.3–2.5	0.0005	0.0006–0.001
LLDPE	2.09	2.2	0.0006	0.0003
LDPE	2.3	2.3–2.45	0.0004	0.0005
PP	2.29	2.213	0.0004	0.00043
Nylon	3.2	3.098–3.215	0.0355	0.018–0.023
TPV	3.11	2.5–3	0.0665	0.05

**Table 2 polymers-13-02658-t002:** Comparison with the bulk and Z5R5 model.

	Dielectric Constant		Loss Tangent	
	Bulk	Z5R5	Bulk	Z5R5
HDPE	2.27	2.27	0.0008	0.0008
LLDPE	2.09	2.09	0.001	0.0095
LDPE	2.2	2.21	0.0007	0.0007
PP	2.27	2.25	0.0005	0.0005
Nylon	3.2	3.18	0.0355	0.041
TPV	3.11	3.13	0.0665	0.066

## Data Availability

The data presented in this study are available on request from the corresponding author.

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
