# Peer review of "Measuring the Complex Permittivities of Plastics in Irregular Shapes"

_polymers, 2021, doi:10.3390/polym13162658_

Round 1
Reviewer 1 Report
In this manuscript, the authors present a measurement of the complex permittivities of six plastic materials in irregular shapes at the microwave frequency, in which the results were evaluated by the simulation and were compared to their counterparts in bulk form. The results are meaningful and important. There are only two minor problems I should addressed.
- The figure captions in Fig.6 should be checked, (d)PE in caption while (d)pp in Figure.
- Why do the authors choose such six plastic materials in this work? Why not the other polymer materials?
Author Response
Our responses can be found in the attached file.

Reviewer 2 Report
This paper reports a method to separate different types of plastic such as LDPE (low density polyethylene), LLDPE (linear low density polyethylene), HDPE (high density polyethylene), and PP (polypropylene). Bulk plastics were shredded with a pencil sharpener and were pushed in sample cells of Teflon. Complex permittivity of the samples were measured by an enhanced-field method.
Volume percentages of the samples were from 10% to 50%. Calculations of a high frequency simulation software (HFSS) were performed to estimate parameters related to permittivity of samples with gaps. An optimal model with layers and rings in the cell was proposed successfully.
The experiments and simulations seem to have been competently conducted. However, I do not understand the need to do the calculations because the volume dependence can be estimate the results from bulk samples and empty cells linearly. The generality of the model has not been fully discussed in conditions of shredded size, shape and size distribution.
Author Response

(The authors gave the same response as above.)
